# Relational Representations Mitigation of Catastrophic Forgetting: Graph Attention Networks for Online Continual Learning for Image Classification

## Abstract

Online Continual Learning (OCL) aims to enable models to learn from non-stationary data streams while preserving knowledge acquired from previously observed tasks. A fundamental challenge in OCL for image classification is catastrophic forgetting. Existing approaches predominantly rely on Convolutional Neural Networks (CNNs), whose grid-based representations of images emphasize local features but insufficiently model relational structures that may be re-used and built upon even under distribution shifts. In this work, we propose ReReM, a Relational Representations Mitigation framework for OCL for image classification that integrates Graph Attention Networks (GATs) with hierarchical features extracted from pretrained vision backbones. Images are transformed into multi-scale graphs, allowing models to learn interactions between semantic regions and enabling continual learning through attention-based message passing over relational structures. This design allows the model to selectively learn and update learned contextually relevant components for emphasis. To improve graph-level aggregation, we introduce a learnable weighted global pooling mechanism that adaptively prioritizes nodes. Furthermore, we propose a rehearsal duplication strategy that re-balances the influence of past and current data during training, improving knowledge retention without increasing memory storage requirements. Extensive experiments on CIFAR10, CIFAR100, and MiniImageNet demonstrate consistent improvements over existing pretrained CNN-based continual learning methods. Additional evaluations using Vision Transformer (ViT) feature extractors further show that the proposed framework generalizes across backbone architectures. Our results suggest that relational graph representations provide an effective inductive bias for mitigating forgetting in online continual learning for image classification.

## 1 Introduction

Online Continual Learning (OCL) for image classification seeks to enable machine learning models to learn from a sequence of tasks presented over time while retaining performance on previously acquired knowledge (Wang et al., 2024; Huo et al., 2024). This learning paradigm is essential for real-world applications operating under non-stationary data distributions, such as autonomous driving and robotic navigation, where models must continuously adapt to new information without retraining from scratch. A central challenge in OCL is *catastrophic forgetting*, whereby learning new tasks interferes with and degrades representations learned from earlier tasks (McCloskey & Cohen, 1989; van de Ven & Tolias, 2019).

Most existing approaches to continual learning for image classification rely on Convolutional Neural Networks (CNNs), which learn hierarchical feature representations through localized spatial aggregation (Jung et al., 2023; van de Ven & Tolias, 2019). While highly effective for static learning settings, CNNs inherently process images as fixed grids, emphasizing local appearance patterns rather than relationships between semantic components. As tasks evolve, such instance-centric representations may change substantially, contributing to instability across sequential learning stages and exacerbating forgetting.

In contrast, Graph Neural Networks (GNNs) provide a natural mechanism for modeling relationships among entities through message passing over structured representations. By representing images as graphs whose nodes correspond to semantic regions and edges representing adjacency for learning contextual interactions, GNNs enable learning based on relational structure rather than solely local features. This perspective aligns with cognitive theories such as Recognition-by-Components (RBC) (Biederman, 1987), where object understanding arises from both component identification and their structural relationships.

Motivated by this observation, we propose ReReM, a novel Relational Representations Mitigation framework for online continual learning that integrates hierarchical visual representations with Graph Attention Networks (GATs). ReReM constructs graphs from multi-scale feature maps extracted by pretrained backbones, including CNNs and Vision Transformers (ViTs), enabling joint modeling of fine-grained and high-level semantic information. Attention-based message passing allows the model to dynamically emphasize informative relationships between regions, producing representations that adapt to new tasks while preserving previously learned structure.

To further improve graph-level representation learning, we introduce a new learnable weighted global pooling mechanism that adaptively prioritizes informative nodes during aggregation, overcoming limitations of conventional pooling strategies that treat all nodes uniformly. In addition, we address the imbalance between current-task data and limited rehearsal memory in continual learning by proposing a rehearsal duplication strategy that increases exposure to past-task samples without expanding memory storage, thereby improving knowledge retention under fixed resource constraints.

Our key contributions are summarized as follows:

1. We introduce a relational representation mitigation framework for online continual learning that combines hierarchical visual features with Graph Attention Networks to model interactions between semantic components of images.

2. We propose a learnable weighted global mean pooling mechanism that enhances graph-level representations by adaptively emphasizing informative nodes during aggregation.

3. We present a rehearsal duplication strategy that rebalances the influence of past and current data during training, improving stability without increasing rehearsal memory requirements.

4. Extensive experiments on CIFAR10, CIFAR100, and MiniImageNet demonstrate consistent improvements over pretrained CNN-based continual learning methods, with additional preliminary validation on Vision Transformer backbones highlighting the generality of the proposed framework.

The remainder of this paper is organized as follows. Section 2 reviews related work for continual learning in image classification, continual learning with GNNs as well as continual learning with rehearsal. Section 3 presents the proposed relational continual learning framework, including hierarchical graph construction, attention-based learning, and rehearsal duplication. Section 4 describes the experimental setup and empirical results, followed by ablation studies and analysis. Finally, Section 5 concludes the paper.

## 2 Related Work

### 2.1 Continual Learning in Image Classification

Continual Learning (CL) is a paradigm designed to enable models to sequentially learn a series of tasks while preserving performance on previously learned tasks, mitigating catastrophic forgetting, a common issue in standard training approaches where new learning overwrites prior knowledge. CL is typically categorized into three main settings which are Task-Incremental, Domain-Incremental, and Class-Incremental (van de Ven & Tolias, 2019; De Lange et al., 2022; van de Ven et al., 2022). Among these, Class-Incremental Learning is particularly challenging, as it requires the model to perform without explicit task identifiers and to classify across all encountered classes, thereby minimizing reliance on task-specific information.

In CL, pretrained models are often employed as feature extractors, leveraging representations generated by the final feature map. These approaches can be broadly divided into two categories: those that treat the pretrained model as a trainable backbone and those that use it as a fixed feature extractor (Jung et al., 2023). Recent work, such as MuFAN, further leverages multi-granularity feature representations by utilizing feature maps from multiple layers to create richer representations for online continual learning (Jung et al., 2023). While one possibility is to directly utilize this representation for classification, our work employs GNNs to advance this concept further. By utilizing GNNs, we enable learning of relationships between features for representation update.

## 2.2 Continual Learning with Graph Neural Networks

GNNs have attracted considerable attention in the field of CL due to their effectiveness in handling graph-based data and enablement of methods to use graph specific information. For example, TWP (Liu et al., 2020) leverages topological information to stabilize training by measuring the gradient of attention coefficients in a GAT. However, most existing studies primarily focus on graph-structured data, making them unsuitable for image classification.

Recent research works have explored bridging the gap to utilize GNNs for CL in image classification. DGN (Carta et al., 2021) employs a static clustering method and evaluates its performance on MNIST and CIFAR10 datasets. Similarly, GCL (Tang & Matteson, 2021) models image batches using random graphs, departing from conventional image-based methods. However, to the best of our knowledge, this is the first work to explore the use of graph attention networks on a hierarchical graph representation of images for CL for image classification.

## 2.3 Continual Learning with Rehearsal

The CL setup restricts the complete storage of past tasks for joint training, requiring models to learn tasks sequentially. Rehearsal-based techniques store a small subset of past tasks in a rehearsal buffer, and joint-training is then emulated through the use of both the current task and rehearsal buffer (Rolnick et al., 2018; Zhou & Cao, 2021). To better emulate joint training, methods operate on the rehearsal buffer, such as ER-GNN (Zhou & Cao, 2021) which selects from the replay buffer with replacement during training and rehearsal example selection based on class clusters and coverage metrics. However, we demonstrate that a straightforward storage and duplication strategy is capable for guiding model training and obtains competitive results.

# 3 Method

## 3.1 Background

**Graph Neural Networks.** A fundamental characteristic of GNNs is the structure of their layers, which directly influences the primary operation of GNNs: the message passing and update step. A common family of layers within GNNs is graph convolutional layers. In their simplest form, these layers can be expressed as:

$$Node' = Aggregate(Neighbour(Node)) \times Node \tag{1}$$

which simply aggregates the information from neighboring nodes into the node being updated. Trainable types of layers generally evolve from this basic formula, such as GraphConv (Morris et al., 2018), which adds a set of trainable weights for both the aggregate as well as node to be updated. Another component that can be swapped out would be the aggregation operation. Switching out the summation operation in GraphConv (Morris et al., 2018) to a mean operation would yield the general update formula used for SageConv (Hamilton et al., 2017). Instead of changing the aggregation operation, the weighing of neighboring nodes can be adjusted as well. GCNConv (Kipf & Welling, 2017) iterates on the general formulation by including the degree of the nodes. GATConv (Veličković et al., 2018) improves on the neighbor nodes weightage by adding learnable attention mechanisms between the neighboring nodes and the node to be updated, as well as self-attention for the node to be updated. GATv2Conv (Brody et al., 2022) further

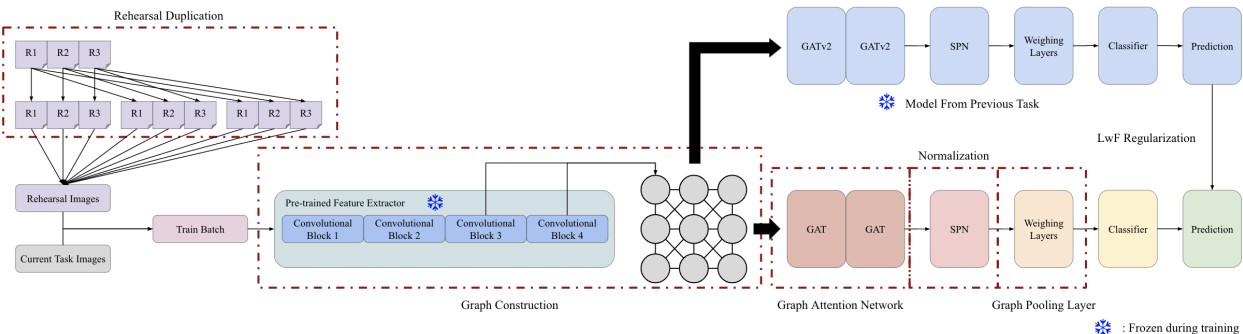

Figure 1: Overview of ReReM, illustrated with three rehearsal images for Rehearsal Duplication. Train batches are first converted into graphs through Hierarchical Graph Construction. A Graph Attention Network is then utilized, followed by a Normalization and a Graph Pooling Layer.

improves upon GATConv by re-arranging the order of operations to allow the attention mechanism to perform better based on the inputs.

### 3.2 Preliminaries

Given $T$ total tasks, the image classes $C = \{c_1, ..., c_\alpha, ...\}$ can be divided into $T$ tasks, denoted as $T = \{t_1, ...\} = \{\{c_1, ...\}, \{c_\alpha, ...\}, ...\}$, where each image associated with a specific task is represented as $i_\beta \in I_t$, $i_\beta = \{x_\beta, y_\beta\}$ such that $y_\beta = c_\beta \in t_\gamma$. We overload the notation $t$ to present both the task and task number, and similarly $i$ to represent both the image and image number. The graph construction step is represented as $f$ while the GNN and pooling + classifier blocks trained till task $t$ is represented as $g_t$ and $m_t$ respectively, while the rehearsal memory accumulated up to task $t$ is denoted as $R_t$.

### 3.3 Graph Attention Network for Image-Based Continual Learning

We present an overview of the proposed ReReM framework in Figure 1, demonstrating how we first perform rehearsal duplication to enhance the representation of past tasks. Train batches are then passed into a Hierarchical Graph Construction step, which utilizes a pre-trained feature extractor to construct a multi-scale graph representation. The graph is then operated on by a Graph Attention Network, followed by Normalization and a novel Graph Pooling layer for graph level classification.

**Hierarchical Graph Construction.** Images are naturally structured as grid-based data rather than graph-based, making direct graph representation challenging. A naive approach for converting an image into a graph involves treating each pixel as a node and establishing edges between adjacent pixels. However, this method is highly inefficient, as it redundantly represents neighboring pixels with similar attributes as separate nodes.

To address this inefficiency, one avenue would be to employ SuperPixel methods such as Simple Linear Iterative Clustering (SLIC) to aggregate neighborhood information into nodes, representing a spatial region rather than individual pixels (Achanta et al., 2010). Although this approach reduces redundancy, it primarily captures color and spatial information.

Instead, we utilize a pretrained feature extractor to obtain a representation that is more information-rich by mapping the image into a feature space. In addition, instead of just utilizing the final output, features from different layers of the pre-trained feature extractor can be incorporated, integrating both fine-grained information with high-level semantic information. This form of hierarchical representation through a multi-level feature aggregation strategy enhances plasticity for OCL (Jung et al., 2023).

When aligning feature maps of differing spatial resolutions, we deliberately avoid the use of linear or bilinear upsampling methods, as such techniques implicitly assume smooth transitions between feature values that

may not correspond to semantic relationships. For example, a feature map encoding the presence of a circular object in one spatial region and its absence in another does not imply a gradual transition between these regions at finer resolutions. Rather, each pixel contributes independently to the resulting representation. Consequently, we instead upsample feature maps by replicating each feature value according to the ratio of spatial resolutions, thereby preserving semantic consistency and preventing the introduction of spurious interpolations. As an illustrative example, upsampling a feature map of (1, 4) yields a representation of (1, 1, 4, 4).

Node feature assignment is performed by associating each node with its corresponding spatial location across the hierarchical feature maps. Specifically, for each node, we concatenate feature vectors extracted from all aligned positions in the multi-scale feature maps, along with the explicit spatial coordinates (X,Y). This procedure produces a rich node representation that jointly encodes multi-granular semantic information and spatial information. Following the approach of SLIC (Achanta et al., 2010), we utilize a k-nearest neighbor (k-NN) approach based on the spatial information to construct edges between nodes based on spatial proximity. This method for constructing edges enables each node to be updated in the context of its neighborhood and surrounding features.

Through these steps, we establish a methodology for constructing a standardized graph structure to serve as input for the subsequent stage of the pipeline containing learnable components.

**Graph Attention Network.** We utilize a GAT to operate on the constructed graph during node updates to incorporate a learnable attention mechanism into the update process. We posit that the inclusion of attention is critical for alleviating catastrophic forgetting in continual learning settings, as it enables the model to selectively emphasize relevant neighboring nodes during message passing, thereby reducing interference from less informative or redundant neighbors.

To do so, in this paper, we propose to use GATv2 (Brody et al., 2022) for learning the feature graphs. GATv2 (Brody et al., 2022) computes the normalized importance between nodes i and j as follows:

$$\alpha_{i,j} = \text{softmax}(a^\top \text{LeakyReLU}(w \cdot [h_i | h_j])) \tag{2}$$

Here, $h_i$ and $h_j$ represent the node features for nodes $i$ and $j$ respectively, where $a$ and $w$ are learnable weights. The node representation is then updated:

$$h_i' = \sigma(\sum_{j \in N_i} \alpha_{ij} \cdot wh_j) \tag{3}$$

whereby $\sigma$ represents a non-linearity operation. This dynamic attention mechanism allows nodes to aggregate information from their neighbors based on learned importance weights.

**Normalization.** For normalization, we extend the hybrid SPN normalization block parameterized by a set of learnable parameters proposed in Jung et al. (2023), which was initially designed to normalize batch and feature dimensions in feature maps from CNNs. We modify the normalization block to operate on GNNs operating on graphs derived from feature map representations, such as the proposed graph construction and GAT-based architecture. To do so, we treat the entire graph as an analogue of a feature map. In this formulation, nodes correspond to pixel locations, while node channels correspond to feature values across the aggregated feature maps.

**Graph Pooling Layer.** Following the update of the graph representation, the final step for image classification is to perform graph-level classification. To serve as input to the classification component of the architecture, the graph is flattened and transformed into a vector. A naïve option, illustrated in Figure 2 (left), is to directly use the full graph representation. However, this results in a sizable input and imposes the constraint of a fixed graph size.

To address these limitations, global pooling operations can be utilized to reduce the the number of nodes by aggregating node representations across the entire graph into a single representative node, as depicted in Figure 2 (right). Common pooling strategies include max pooling, as adopted in DGN (Carta et al., 2021), as well as addition and mean pooling operations (Fey & Lenssen, 2019).

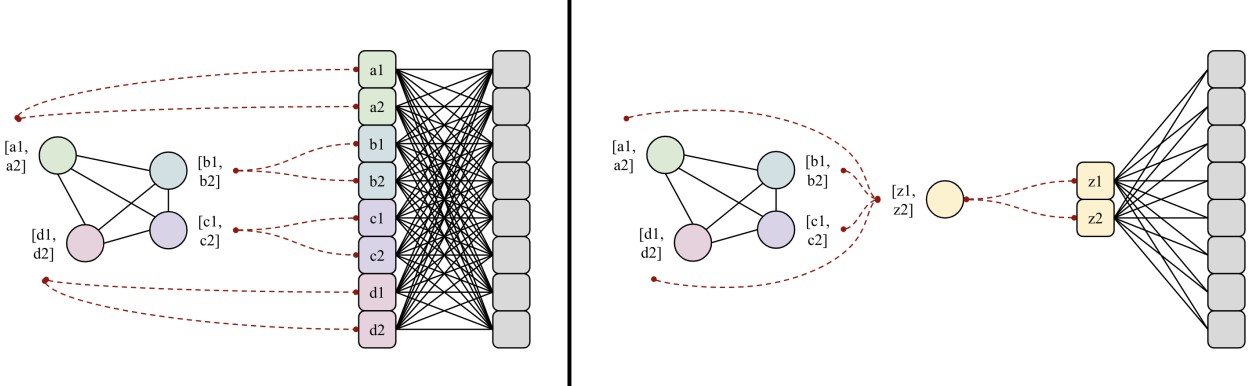

Figure 2: Utilization of entire graph for classification (left), utilization of global pooling before classification (right)

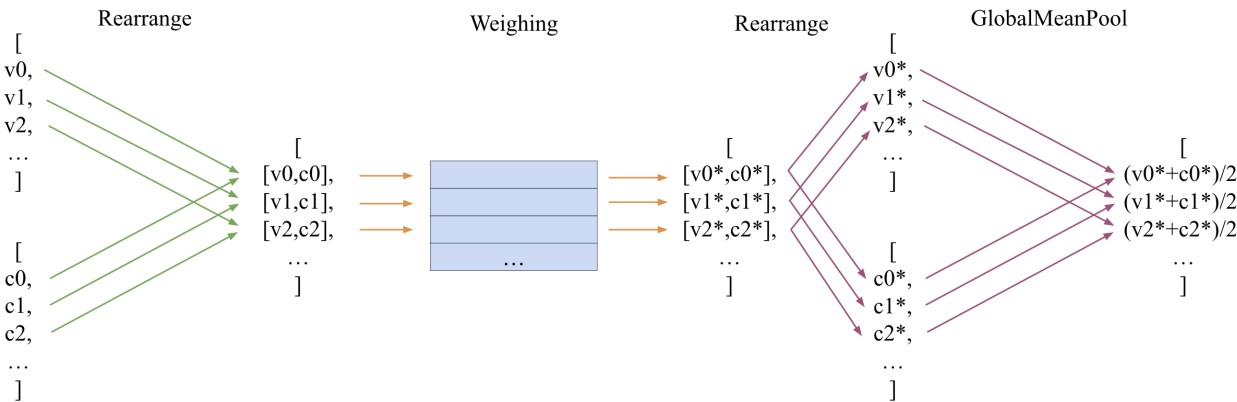

Figure 3: Process of performing weighted global mean pool

Although global mean pooling enables all nodes to contribute to the final representation, thereby capturing information from across the entire graph, it inherently assigns equal importance to all nodes. This uniform weighing can be problematic in the presence of repetitive background structures, which may dominate the aggregated representation.

Additionally, while some methods treat the fixed length embedding as a limitation, developing methods with dynamic adaptation (Li et al., 2024), we treat this property as desirable, allowing the usage of graphs of any size and by extension images of various sizes and therefore choose to expand on the base formulation. Furthermore, alternative methods such as SortPool (Zhang et al., 2018), SoPool (Wang & Ji, 2020) and MLAP (Itoh et al., 2021) are developed primarily on graph based datasets.

To tackle the above, we introduce a image first, learnable channel-wise weighted pooling strategy designed for image derived graphs, building on the Global Mean Pool method. As illustrated in Figure 3, we first reshape the graph representation such that values corresponding to the same channel across different nodes are aligned along a common dimension to enable joint processing. A linear layer is then applied to independently weigh each channel, after which the representation is reshaped back to its original graph format to produce a weighted graph representation. Finally, mean pooling is applied to obtain the graph-level representation for classification.

This weighing scheme preserves the relative importance of channel values across nodes in the final representation. By adaptively weighing each node's channel-wise contribution prior to pooling, the model can

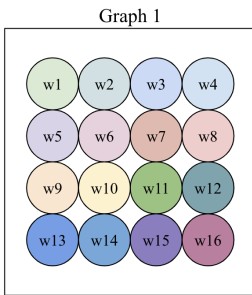 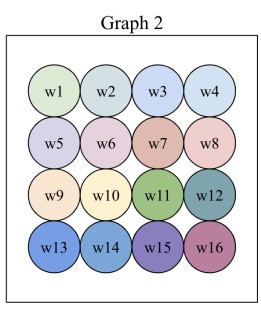 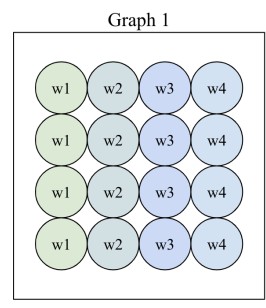 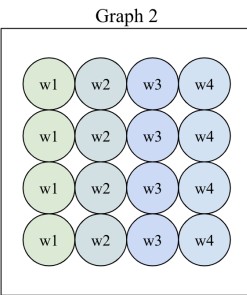

Figure 4: When weighing the entire graph at once (left), each node in the graph receives a separate weight (w1-w16) when passed through the weighing layer. In the alternative form that operates on subsets (right), subsets of the graph, in this case, sets of 4 nodes receives separate weights (w1-w4), which are shared across the rest of the graph.

emphasize key nodes while still allowing information from all nodes to influence the aggregated representation.

While jointly weighing all nodes enables comprehensive comparison across the entire graph, it requires a fixed graph size to determine the size of the weights. To alleviate this constraint, we propose an alternative formulation that operates on smaller subsets of nodes, specifically, partitions comprising one-quarter of the graph at a time. A shared weighing layer is applied independently to each subset, allowing the approach to scale gracefully as the graph size increases in increments of the subset size. Figure 4 illustrates the partitioning procedure and the application of shared weights across subsets. In this toy example, the nodes (represented by the circles) receive different weights (represented by the corresponding w number and the different colors). The weighing layers have $[w1, w2, w3, ..., w16]$ on the left while on the right, the weighing layer only needs $[w1, w2, w3, w4]$.

### 3.4 Training with Rehearsal Duplication

In rehearsal-based continual learning, a limited subset of data from previously encountered tasks is retained in a memory buffer and replayed during subsequent training phases. This replay buffer is typically substantially smaller than the dataset of the current task, leading to an imbalance in representation between past and present tasks during training. To mitigate this issue, we introduce a simple yet effective strategy: *duplicate rehearsal samples prior to their integration into the training stream to better approximate joint-task training.* Concretely, each stored sample is replicated multiple times before integration into the training stream, increasing its frequency of exposure to images from past tasks without expanding the memory buffer. This procedure amplifies the influence of prior-task data during learning and helps preserve previously acquired knowledge while incurring no additional storage cost.

## 4 Experiments

### 4.1 Datasets

We utilize three benchmark datasets for our experiments: 5-Task CIFAR10 (Krizhevsky, 2012; Tang & Matteson, 2021), 20-Task CIFAR100 (Krizhevsky, 2012; Lopez-Paz & Ranzato, 2017), and 20-Task MiniImageNet (Ravi & Larochelle, 2017; Vinyals et al., 2016; Jung et al., 2023).

CIFAR10 (Krizhevsky, 2012) contains 10 image classes with images of size 32x32, and Split CIFAR10 splits the said 10 classes into 5 tasks (Tang & Matteson, 2021). CIFAR100 (Krizhevsky, 2012) contains 100 image classes with images of size 32x32 as well, and Split CIFAR100 splits the mentioned 100 image classes into either 10 tasks (Zhou et al., 2024) or 20 tasks (Lopez-Paz & Ranzato, 2017).

Table 1: Simple Linear Iterative Clustering (SLIC) max nodes

| Dataset | Image Resolution | Max Nodes |
|---|---|---|
| CIFAR100 | 32x32 | 100 |
| MiniImageNet | 84x84 | 700 |

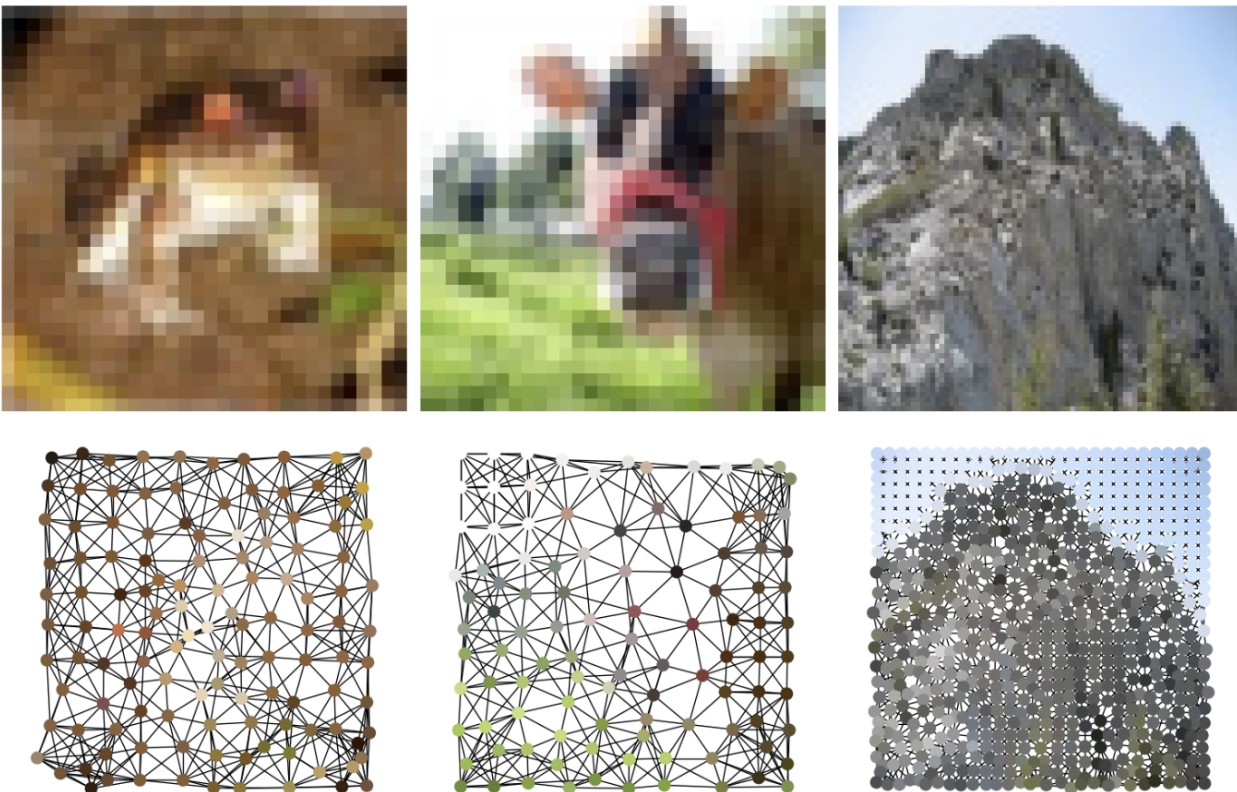

Figure 5: Sample images from, in order, CIFAR10, CIFAR100, MiniImageNet (top) and the corresponding SuperPixel graph (bottom)

While MiniImageNet (Ravi & Larochelle, 2017; Vinyals et al., 2016) also contains 100 image classes, with Split MiniImageNet also dividing the dataset into 20 tasks (Jung et al., 2023), the image resolution is different. The original resolution of the images is 224x224 as the images are derived from the ImageNet dataset. However, instead of utilizing this resolution, to maintain the intended level of image detail, we follow the MiniImageNet process of downscaling the images to a resolution of 84x84 before re-upscaling it to 224x224 as needed for processing.

Through utilization of the aforementioned CL setups, we ensure testing on a varied number of tasks as well as on different datasets which are split into the same number of tasks.

## 4.2 SuperPixel Construction

Where PyTorch Geometric (Fey & Lenssen, 2019) does not have available superpixel versions of datasets generated using SLIC (Achanta et al., 2010), the required data was constructed using the ToSLIC method, and the maximum number of nodes is configured as described in Table 1 with a visual example illustrated in Figure 5.

### 4.3 Architecture

We utilize a frozen pretrained ResNet18 network, stretching smaller feature maps to match the dimensions of the larger feature maps as detailed in Section 3.3: $i \rightarrow e_1 \in \mathbb{R}^{C_1 \times W_1 \times H_1}, e_2 \in \mathbb{R}^{C_2 \times W_2 \times H_2} \rightarrow e_1 \in \mathbb{R}^{C_1 \times W_1 \times H_1}, e_2 \in \mathbb{R}^{C_2 \times W_1 \times H_1} \rightarrow e_{final} \in \mathbb{R}^{(C_1+C_2) \times W_1 \times H_1}$ where W is the width, H is the height and C is the feature map's number of channels. Specifically, output representations from the final two convolutional blocks of the ResNet18 model is used.

Specifically, we utilized the ImageNet-1K (Deng et al., 2009) pre-trained model implemented by PyTorch (Ansel et al., 2024). Prior to re-scaling and combination, we obtain intermediate forms of size $256 \times 14 \times 14$ and $512 \times 7 \times 7$ as we utilize the final 2 convolutional blocks as mentioned. For edge construction using k-NN, we utilize a value of 8 for k following the SLIC (Achanta et al., 2010) approach in Dwivedi et al. (2024) The GNN is implemented utilizing 2 GATv2Conv (Fey & Lenssen, 2019) layers, specifically with 128 channels. The number of attention heads utilized is 4 attention heads for CIFAR10, and 3 attention heads for CIFAR100 and MiniImageNet Normalization is done with the extended SPN normalization block adapted to function on GNNs as detailed in Section 3.3 with reference implementation from the code base made available by MuFAN (Jung et al., 2023). This extended SPN block is used post GNN but before the weighted global mean pooling. For classification, the classifier component has 2 linear layers of 128 features as well. Therefore, the learnable components of the architecture are the GAT layers, normalization parameters, weighted global pooling weights as well as the classifier layers.

For the extended testing by swithing to a ViT (Dosovitskiy et al., 2021) backbone, HuggingFace's "vit-base-patch16-224-in21k" (Wolf et al., 2019; Wu et al., 2020; Deng et al., 2009) was utilized. For a similar hierarchical construction, the representation of size $197 \times 768$ from the final 4 blocks are utilized. As the dimensions of the representation obtained from each block is the same, scaling is not necessary and the outputs can instead be directly concatenated. Furthermore, as the output for ViT is a sequence which differs from the grid obtained from CNNs, to obtain a similar spatial reach, the edge construction is modified to use a k value of 3 instead. When applying this to CIFAR100, we utilize the same number of channels, normalization and classifier setup as described earlier but with 4 attention heads.

### 4.4 Training.

During training, we distill logit information by leveraging the DGN implementation (Carta et al., 2021) of the Learning without Forgetting (LwF) framework (Li & Hoiem, 2016), distilling across the combined training and rehearsal set. Soft targets are obtained by passing each batch from the combined training stream through a saved frozen copy of the model from the previous task. Specifically, the logits distillation loss is formulated as follows:

$$L_{LwF} = \lambda \frac{t^2}{t+1} \sum_{i=0}^{|R_t \cup I_t|} l_{kl}(p_i^{\mathrm{old}}, \log(p_i^{\mathrm{new}})), \tag{4}$$

where the output of the frozen model from the previous task is $p_i^{old}$ and the output of the model from the current task is $p_i^{new}$, i: $p_i = \mathrm{softmax}(m_t(g_t(f(x_i)))/T)$ and $l_{kl}$ represents the Kullback-Leibler divergence. $R_t$ and $I_t$ represents the rehearsal buffer and train set for task $t$ respectively. Logit distillation strength is controlled by a hyper-parameter $\lambda$ and the temperature hyper-parameter is $T$. All results are reported as an average over 5 runs and with LwF as the distillation loss, in addition to standard cross entropy loss for image classification training.

### 4.5 Hyper-parameters

Lambda and temp for LwF was searched using a random search approach, in the range of 0.25 to 2.00 and 1.00 to 3.00 respectively. For the rehearsal budget, we follow GCL (Tang & Matteson, 2021) for CIFAR10 and utilize 5 rehearsal/class which is 10 rehearsal/task. The budget for CIFAR100 and MiniImageNet follows MuFAN (Jung et al., 2023) for 10 rehearsal/class which results in 50 rehearsal/task. When extended to ViT on CIFAR100, the budget is utilize 20 rehearsal/class following EASE (Zhou et al., 2024). The duplication level utilized is set at 15 for CIFAR10 as well as MiniImageNet and at 20 for CIFAR100. Training was

Table 2: Average accuracy ↑ for replicating DGN in OCL across various hyper-parameters

| Num Layer | Lambda | CIFAR10 | CIFAR100 | MiniImageNet |
|-----------|--------|---------|----------|--------------|
| 2 Layers | 0.01 | 13.75±0.75 | 7.09±0.75 | 7.30±1.13 |
| | 0.001 | **14.32±0.30** | 5.49±1.37 | 4.76±0.65 |
| | 0.0001 | 14.11±0.53 | 6.45±0.87 | 6.04±1.77 |
| 4 Layers | 0.01 | 14.00±0.33 | **8.35±0.64** | 7.41±1.84 |
| | 0.001 | 13.83±0.86 | 7.68±0.57 | **7.86±0.53** |
| | 0.0001 | 14.04±0.27 | 7.87±0.62 | 7.43±0.93 |

Table 3: Comparison of average accuracy ↑ with CNN and GNN methods (left), and pretrained feature extractor and GNN methods (right). (**\***) denotes methods that use features from a pretrained feature extractor as input; all others operate directly on raw image pixels.

| Model Family | Method | CIFAR10 | Model Family | Method | CIFAR100 | MiniImageNet |
|--------------|--------|---------|--------------|--------|----------|--------------|
| CNN | EWC | 18.49±0.13 | CNN* | ER | 20.50±0.90 | 11.00±0.50 |
| | GEM | 22.88±4.06 | | DER++ | 20.70±2.70 | 13.70±1.20 |
| | ER | 29.94±2.08 | | DualNet | 25.50±0.70 | 20.90±1.60 |
| | GCL | 49.62±1.85 | | MuFAN | 39.60±0.30 | 34.70±2.10 |
| GNN | DGN | 14.32±0.30 | GNN | DGN | 8.35±0.64 | 7.86±0.53 |
| GNN* | ReReM | **63.07±2.10** | GNN* | ReReM | **43.95±1.16** | **41.20±2.57** |

performed on NVIDIA A100 and RTX A6000 GPUs, utilizing the Adam optimizer and a learning rate of 0.001.

## 4.6 Metrics

Performance is reported primarily using Average Accuracy (Tang & Matteson, 2021), which is defined using $Accuracy_{i,j}$, which is the accuracy of Task j after training till Task i,

$$Average\ Accuracy = \frac{1}{T}\sum_{j=0}^{T} Accuracy_{T,j} \tag{5}$$

This is supplemented with Average Forgetting as defined by GCL (Tang & Matteson, 2021) as well,

$$Average\ Forgetting = \frac{1}{T-1}\sum_{j=0}^{T-1}(Accuracy_{T,j} - Accuracy_{j,j}) \tag{6}$$

## 4.7 DGN Replication

For completeness and to aid transparency and reproducibility, we also report the full results for DGN utilizing SLIC, modified for the online setting across various hyper-parameters in Table 2. We select results from the best performing configuration for comparison in subsequent tables.

## 4.8 Results on Online Class-Incremental Learning

We first present the average accuracy results in Table 3 for an online class-incremental setting, whereby task identities are not given but the model needs to perform classification over all image classes. For the CIFAR10 benchmark, results for CNN based methods are referenced from GCL (Tang & Matteson, 2021) while on CIFAR100 and MiniImageNet, results obtained are compared against the reported performances in MuFAN (Jung et al., 2023). In addition, for GNN comparisons, we include comparison against the best reproduced

Table 4: Comparison of average forgetting ↓ with CNN and GNN methods. (**\***) denotes methods that use features from a pretrained feature extractor as input; all others operate directly on raw image pixels.

| Dataset | CNN | | | | GNN | GNN**\*** |
| | EWC | GEM | ER | GCL | DGN | ReReM |
|---|---|---|---|---|---|---|
| CIFAR10 | 86.95±1.15 | 76.90±5.53 | 72.64±4.88 | 35.69±3.33 | 71.02±0.89 | **24.30±4.27** |

Table 5: Comparison of average accuracy ↑ with the ViT backbone on CIFAR100 with offline 10 Task rehearsal focused methods.

| Dataset | 10 Task Rehearsal | | | ReReM |
| | iCaRL | DER | MEMO | |
|---|---|---|---|---|
| CIFAR100 | 73.87 | 77.93 | 75.79 | **82.55** |

DGN (Carta et al., 2021) values from Table 2. Through this evaluation protocol, we enable a comprehensive comparison against CNNs that both utilize graph representation as well as pre-trained feature extractors as well as GNN based approaches. In this comparison, our method achieves significant improvements in average accuracy over the compared methods, with accuracy gains of 27%, 10% and 18% on CIFAR10, CIFAR100, and MiniImageNet respectively.

In addition, we also compare average forgetting against the results reported by GCL (Tang & Matteson, 2021) for CIFAR10 in Table 4. When evaluating the tabled results, our method outperforms the compared approach on CIFAR10 significantly, which is consistent with the results for the average accuracy metric.

### 4.9 Extension to ViT Backbone

To further explore the proposed method's generalization capability, we conduct a preliminary extension investigation by switching to a Vision Transformer (ViT) (Dosovitskiy et al., 2021), replacing the ResNet18 feature extractor (He et al., 2015), as described in Section 4.3. We evaluate our method against ViT-based offline baselines on the 10-Task split of CIFAR-100 in Table 5, referencing results from Zhou et al. (2024).

The results indicate that the proposed method consistently outperforms the baseline approaches. We highlight that the baseline results reported in Table 5 are obtained from a different offline setting, which is considerably less demanding than the OCL scenario under which our method is evaluated. We contend that the performance of these baselines would likely deteriorate further when deployed in an OCL setting, thereby further highlighting the ability of the proposed approach, performing in a more constrained scenario, regardless of the backbone architectures. From these, we highlight the extension of a ViT hybrid architecture and rigorous comparison with online ViT baselines as key directions for future works.

### 4.10 Ablation Study

We conduct an ablation study to assess the contributions of the proposed attention mechanism and the rehearsal duplication strategy. As reported in Table 6, utilization of GAT for attention-based updates consistently improves average accuracy across the evaluated datasets when compared to non-attention alternatives, such as GraphConv (Morris et al., 2018). Orthogonally, the integration of rehearsal duplication contributes substantial additional performance gains, demonstrating that the representation imbalance issue in OCL can be mitigated with a simple rehearsal replication strategy effectively. However the best variant remains when both GAT and rehearsal duplication is applied, showing the complementary aspects of both components of the framework. Furthermore, we also evaluate a baseline performance by switching the GAT layers to be linear layers instead, to further isolate the performance of the GNN component. We observe that in general, this setup underperforms the full framework, highlighting the benefit of utilizing graphs and the GAT architecture.

Table 6: Ablation across all three datasets for proposed utilization of Graph Attention (G. Attn) and Rehearsal Duplication (R. Duplication), values shown are average accuracy ↑

| Method | CIFAR10 | CIFAR100 | MiniImageNet |
|---|---|---|---|
| GNN | 47.40 | 15.15 | 28.26 |
| GNN + G. Attn | 49.00 | 15.97 | 32.55 |
| GNN + R. Duplication | 60.04 | 42.16 | 39.35 |
| GNN + G. Attn + R. Duplication | **63.07** | **43.95** | **41.20** |
| Baseline + R. Duplication | 62.80 | 39.92 | 39.84 |

Table 7: Ablation across all three datasets for proposed utilization of Weighted Global Mean Pooling (Ours) compared to other global pooling operations available, values shown are average accuracy ↑

| Dataset | CIFAR10 | CIFAR100 | MiniImageNet |
|---|---|---|---|
| GlobalAddPool | 12.40 | 21.10 | 34.70 |
| GlobalMaxPool | 47.45 | 32.15 | 38.60 |
| GlobalMeanPool | 58.36 | 43.69 | 40.66 |
| Ours | **63.07** | **43.95** | **41.20** |

Table 8: Joint training across all three datasets, values shown are accuracy ↑

| Dataset | CIFAR10 | CIFAR100 | MiniImageNet |
|---|---|---|---|
| Baseline | 84.68 | 55.16 | 72.82 |
| GNN + Graph Attention | **85.39** | **57.15** | **74.35** |

We also leverage this Baseline implementation for comparison in a setting without any CL techniques for mitigating catastrophic forgetting, training with only cross-entropy loss and present the results in Table 9. We observe that in most cases, the proposed architecture leveraging graphs slightly outperforms the Baseline setup which does not not utilize the graph structure due to the lack of a GNN. However we also note that in the absence of any CL techniques, catastrophic forgetting is the predominant effect and occurs to a large degree leading to low average accuracy numbers across the board.

In addition, we evaluate the proposed weighted global mean pooling strategy's effectiveness relative to existing global pooling operations, with the results presented in Table 7. The findings indicate that additive pooling underperforms consistently, whereas mean pooling approaches achieves superior results. Notably, the proposed weighted mean pooling method consistently outperforms all alternative pooling strategies across all benchmark datasets, highlighting the advantage of enabling the model to learn node importance during graph-level aggregation.

We also utilize the Baseline implementation for comparison in a joint training setting to study if the observed performance gains in a CL setting is also applicable in overall performance and present the results in Table 8. We observe that across the board, the proposed architecture outperforms the Baseline, hinting that the benefits are generalizable and not just isolated to the CL setting. Furthermore, we observe that for CIFAR100, the improvements for the proposed architecture is greater in the CL scenario (+4.03) in Table 6 as compared to joint training (+1.99), indicating that the performance gain goes beyond the improved upper bound and that the proposed architecture is also beneficial to CL specifically.

## 5 Conclusion

In this work, we presented a novel OCL framework for image classification that leverages Graph Attention Networks (GATs) to effectively model contextual relationships and dynamically update class-specific representations. The proposed approach constructs information-rich graph representations by integrating

Table 9: Training without CL techniques across all three datasets, values shown are average accuracy ↑

| Dataset | CIFAR10 | CIFAR100 | MiniImageNet |
|---|---|---|---|
| Baseline | 19.21 | **2.65** | 3.22 |
| GNN + Graph Attention | **19.31** | 2.62 | **3.32** |

multi-scale semantic information obtained from a pretrained backbone in the hierarchical graph construction process. The GAT layers then operate on this graph representation, performing relational message passing and updating the graph representation. To enhance graph-level classification, we introduced a weighted global pooling mechanism that learns and selectively emphasizes key nodes during pooling. Lastly, we proposed a simple yet effective rehearsal duplication strategy to mitigate representational imbalance between current and previously learned tasks without increasing memory requirements, improving retention of past-task knowledge.

Extensive experiments conducted on three standard benchmark datasets demonstrate that the proposed method consistently outperforms pretrained CNN-based approaches, underscoring its effectiveness in OCL scenarios. Through ablation studies, we reveal that the components of the framework consistently improves performance, complementing one another, with rehearsal duplication having the most significant contribution, and the attention based mechanism and replay stabilization mechanism providing orthogonal and complementary benefits.

Future work could extend upon the proposed work by further leveraging the components of the described framework to construct specific techniques that better leverage the framework. Examples of such potential avenues could be graph and graph attention related techniques to further leverage the GNN component of the framework. While this framework focused on a hierarchical and thus grid-aligned construction with consistent node counts, further investigation into the application and mapping of the developed framework onto non-grid aligned graphs with varying node counts, such as those generated by SLIC would additionally extend the method's applicability and utility.

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
