# OpenReview forum: "Relational Representations Mitigation of Catastrophic Forgetting: Graph Attention Networks for Online Continual Learning for Image Classification"
_TMLR — Rejected by TMLR_

### Review · Reviewer_f1WV · 2026-04-28

**Summary Of Contributions:**

This paper argues that building a semantically richer graph representation of images on top of pretrained feature extractors, and learning over it with attention-based message passing, provides a better inductive bias for CL. They show that combined with LwF distillation and a rehearsal duplication strategy to rebalance past and present task data, this approach effectively mitigates catastrophic forgetting in computer vision tasks.

More specifically the contributions of the paper are:

* arguing that an attention-based graph representation capturing semantic information of images is useful against catastrophic forgetting in online CL computer vision tasks
* proposing a framework to mitigate forgetting in computer vision CL tasks by: using a multi-scale graph representation of images, learning a global pooling mechanism of this representation to perform the downstream task and optimising the use of stored past data to prevent forgetting via rehearsal duplication and LwF distillation
* showing that the proposed framework leads to reduced forgetting compared to approaches that directly use CNN or GNN feature representations for classification

Strengths:

* the paper is clear, succinct and easy to understand and follow
* the core idea is interesting, and departs from the standard strategy of developing new ad-hoc CL algorithms for the specific task at hand, focusing instead on alternative architectures that provide better inductive bias for CL
* the results show effectiveness of the method and ablate components of the approach to show their relative contributions

Weaknesses:

* the motivation on why using graph networks and attention-based message passing for handling images is intuitive but a bit vague and not directly supported by the results - do you expect this to only apply to computer vision tasks where semantics are key? what about tasks where low-level feature extraction is the objective?
* results could be made stronger by adding certain baselines (see requested changes, e.g. a baseline without LwF and rehearsal protection to evaluate the intrinsic effect of the graph+attention strategy in CL)
* in certain result tables (e.g. tables 6 and 7) the differences in the results are small, making it difficult to draw final conclusions without random seed error bounds

**Audience:**

Yes

**Audience Explanation:**

Yes, this paper discusses an alternative way to mitigate catastrophic forgetting in computer vision tasks which could be of interest to the community: instead of providing a new algorithmic approach for CL, the paper argues to use a smarter handing of semantic information via graph neural networks and attention-based message passing so that catastrophic interference is mitigated.

**Broader Impact Concerns:**

None.

**Claims And Evidence:**

Yes

**Claims Explanation:**

Yes. The paper provides results across a set of computer vision benchmarks and a set of ablation experiments to demonstrate the utility of different components of the approach. Importantly though, the title/abstract should be refined to make sure that results only support the claim on computer vision tasks, and not for online continual learning in general.

**Requested Changes:**

* Openly acknowledge that current results apply to CL settings for computer vision tasks only: at the moment the title and abstract seem to imply that the results apply across modalities, not just images
* There is no mention to related work on learnable pooling mechanisms for GNNs. Can you add this?
* Experimental section could be made stronger by adding:
    * results on using graph representation without any CL protection - do these results support the hypothesis that a graph structure mitigates forgetting intrinsically compared to a CNN backbone?
    * multitask results (i.e. training on all tasks simultaneously) to separate the impact of using a graph neural network in 1) the overall performance and 2) the performance in a CL setting


* Improve section 3.3 by being more explicit about:
    * whether the graph structure of edges/nodes is shared across tasks? if so, which task data is used to derive the graph?
    * which parameters are exactly learned sequentially across tasks? attention weights of graph and global pooling coefficients?
* Figure 4 is a little cryptic at the moment and hard to understand, either remove or clarify further.
* Improve section 4.4 by making sure all variables in Eq (4) are introduced, or reminding what they are even if they were introduced much earlier in the text? Clarify if this is this the total loss?
* It is not clear to me why Table 2 interesting? There is no discussion on it at the moment.
* Table 3 has weird formatting, making it hard to parse. Can you make a single Model Family - Method - CIFAR-10 - CIFAR100- MiniImageNet table and fill in the rows that are available?
* Table 5, be more explicit in the caption that this is offline?
* In table 6 and 7, be explicit that you are showing accuracies.

---

> ### Author Response · Authors · 2026-05-19
> **Response to Reviewer f1WV (1/2)**
>
> We thank the reviewer for the thorough reading and comprehensive feedback provided. We address the requested changes as follows:
>
> 1. We thank the reviewer for this important observation and agree that the original wording in the title and abstract could unintentionally suggest broader applicability beyond the evaluated domain of computer vision.
>
>    &nbsp;
>
>    To address this, we have revised the title to explicitly reference online continual learning for image classification and have updated the abstract to consistently contextualize ReReM within image-based continual learning settings.
>
>    &nbsp;
>
> 2. We thank the reviewer for this helpful suggestion. We have expanded Section 3.3 to include discussion of related learnable pooling approaches for GNNs, including methods such as SortPool, SoPool, and MLAP. We further clarify how the proposed weighted global mean pooling differs from these approaches, being developed as an image-first approach with the goal of preserving scalability and flexibility for varying graph sizes.
>
>    &nbsp;
>
> 3. We thank the reviewer for these insightful suggestions. To strengthen the experimental analysis, we have added both sets of experiments requested.
>
>    &nbsp;
>
>    First, we included experiments without continual learning mitigation techniques (Table 9), where models are trained using only the standard cross-entropy objective without replay or distillation. These experiments help isolate whether graph-based relational representations themselves provide intrinsic robustness against catastrophic forgetting. The results show that the graph-based formulation achieves slight improvements over the non-graph baseline in most cases, suggesting that relational graph structures might provide a beneficial inductive bias even in the absence of explicit CL protection mechanisms, although catastrophic forgetting still remains substantial overall due to the lack of mitigating techniques.
>
>    &nbsp;
>
>    Second, we added multitask joint-training experiments (Table 8), where all tasks are trained simultaneously. These results help disentangle the benefits of the proposed graph-based architecture in terms of overall representation learning capacity versus continual learning-specific robustness. We observe that the graph-based formulation consistently improves performance even in the multitask setting, indicating that the proposed architecture contributes to stronger overall representations. Furthermore, in the CIFAR100 setup, the relative gains are much larger in the continual learning setting than in multitask training, suggesting that the benefits extend beyond improved upper-bound performance and are particularly relevant to mitigating forgetting under sequential learning.
>
>    &nbsp;
>
>    We have updated the experimental discussion accordingly to better contextualize these findings and clarify the role of graph-based relational learning in both standard and continual learning settings.
>
>    &nbsp;
>
>    For clarity during the rebuttal phase, Table 9 is currently placed later in the manuscript to preserve the existing table numbering and discussion flow referenced throughout the review process. It will be repositioned appropriately in the final revised version.
>
>    &nbsp;
>
> 4. We thank the reviewer for this helpful suggestion. We have revised Section 3.3 to explicitly describe the graph construction process as a standardized approach and highlight the learnable components of the framework. Specifically, the graph construction procedure itself is shared consistently across all tasks: node definitions are derived from the hierarchical feature maps extracted by the fixed pretrained backbone, while edges are constructed dynamically for each input image using the same k-NN spatial connectivity strategy. Thus, the graph construction rule remains fixed across tasks, while the instantiated graphs depend on the input image being processed rather than on any specific task-level graph template.
>
>    &nbsp;
>
>    We further clarified in Section 4.3 that the parameters learned include the GAT layers, the normalization parameters, the weighted global pooling weights, and the classifier layers. These components are updated continuously throughout sequential training using the combined current-task and rehearsal data streams.
>
>    &nbsp;
>
> 5. We thank the reviewer for this feedback and agree that the original presentation of Figure 4 could be clearer.
>
>    &nbsp;
>
>    To improve readability and interpretation, we have revised both the figure caption and the surrounding discussion in Section 3.3 to provide a more explicit explanation of the weighing process and the distinction between the full-graph and subset-based formulation. In particular, we now clarify how weights are assigned across nodes and how shared weighing operates in the subset formulation.

---

> > ### Author Response · Authors · 2026-05-19
> > **Response to Reviewer f1WV (2/2)**
> >
> > 6. We thank the reviewer for this helpful suggestion. To improve readability, we have revised Section 4.4 to include brief in-line reminders for the variables appearing in Eq. (4), even when they were introduced earlier in the manuscript. In particular, we now explicitly restate the meanings of the rehearsal buffer and current-task dataset directly around the equation for easier reference.
> >
> >    &nbsp;
> >
> >    We have also clarified that Eq. (4) represents the LwF distillation component of the objective and that the overall training loss consists of this distillation loss in addition to the standard cross-entropy classification loss used for image classification training.
> >
> >    &nbsp;
> >
> > 7. We thank the reviewer for this observation. The intention of Table 2 is to document the replication and hyperparameter exploration process for the DGN baseline used in our comparisons. Since the original DGN work was not designed for the exact online continual learning setting evaluated in this paper, we performed a reproduction study across multiple hyperparameter configurations to identify the strongest-performing variant under our experimental setup. The best-performing configurations from Table 2 are then used as the DGN comparison baseline throughout the remainder of the paper.
> >
> >    &nbsp;
> >
> >    To clarify this purpose, we have revised the discussion surrounding Table 2 to explicitly state that it is included for transparency and reproducibility of the reproduced DGN baseline rather than as a standalone experimental contribution. We have also improved the surrounding text to better connect the table to the subsequent comparative evaluations.
> >
> >    &nbsp;
> >
> > 8. We thank the reviewer for this feedback and agree that the original formatting of Table 3 could make the comparisons harder to parse visually.
> >
> >    &nbsp;
> >
> >    Our intention in separating the table was to distinguish between different comparison settings and to more clearly group methods according to the type of backbone and representation utilized, while also preserving attribution consistency with the respective referenced works. However, we acknowledge that this presentation may reduce readability.
> >
> >    &nbsp;
> >
> >    To improve clarity, we have revised the formatting of Table 3 by increasing the spacing between the two comparison groups, adding clearer visual separation, and introducing a vertical divider to better distinguish the sub-tables.
> >
> >    &nbsp;
> >
> > 9. We thank the reviewer for this helpful suggestion. To improve clarity and avoid potential misinterpretation, we have revised the caption and surrounding discussion to explicitly state that the referenced comparison methods operate in offline rehearsal settings, whereas ReReM is evaluated under the more constrained online continual learning scenario. This clarification is intended to ensure that readers interpret the comparison appropriately and understand that the table is presented primarily as a preliminary backbone generalization study rather than a strictly matched benchmarking comparison.
> >
> >    &nbsp;
> >
> > 10. We thank the reviewer for this helpful suggestion. We agree that the captions for Tables 6 and 7 should more explicitly specify the reported metric for improved readability and consistency. To address this, we have revised the table captions to explicitly state that the reported values correspond to average accuracy results.

---

> > > ### Comment · Reviewer_f1WV · 2026-05-26
> > >
> > > I thank the authors for the detailed reply and for addressing my comments.
> > >
> > > Based on the latest provided results I have some new concerns. Specifically, table 6 now clearly shows that rehearsal duplication is clearly the main driver of improvement in the CL setting, and that a non-GNN backbone with rehearsal performs quite closely to the full GNN + G. Attn + R. Duplication. Given that GNN + G. Attn result in non-trivial performance, I still believe the results are interesting. However, I think in light of the new results I believe there is still an overstatement of contributions.
> > >
> > > For example, given how crucial rehearsal is, I am uncomfortable with the emphasis put on the "rehearsal duplication" contribution. How different is this rehearsal to other forms of rehearsal? Is this not very close to rehearsal with replacement, which has already been previously done? I believe a deeper discussion on this topic is missing in the paper given the current framing.

---

### Review · Reviewer_tg3v · 2026-04-30

**Summary Of Contributions:**

The paper introduces a novel framework called the *"Relational Representations Mitigation"* framework that claims to improve online continual training for images. The framework has been benchmarked on three datasets—CIFAR-10, CIFAR-100, and MiniImageNet—and shows improved performance in terms of accuracy and forgetting compared to other baselines. The framework achieves this through four components:

1. Hierarchical graph construction from multi-scale feature maps
2. Graph Attention Networks
3. Weighted global pooling
4. Rehearsal duplication strategy

## Strengths

1. The improvements in performance are substantial across all three datasets.
2. The motivation behind each of the four components is well justified.

## Weaknesses

1. The component analysis (Tables 6 and 7) is not sufficiently elaborate for the rehearsal duplication strategy and hierarchical graph construction.
2. The authors mention superpixel construction using SLIC as an experiment (Section 4.2). This is misleading, as SLIC is not part of the main pipeline.
3. The naming of the framework as *"Relational Representations Mitigation"* is not entirely clear. The name primarily reflects only the first component (hierarchical graph construction), while the contribution of that component is not thoroughly studied.

**Additional Comments:**

n/A

**Audience:**

Yes

**Audience Explanation:**

The simple rehearsal duplication strategy proposed by the paper appears to be a plug-and-play addition to any existing framework on OCL for any modality of data. Hence, the paper must be of interest and use to researchers working on continual learning.

**Claims And Evidence:**

No

**Claims Explanation:**

The framework, while demonstrating substantial performance improvements over the baselines, does not clearly attribute these gains to specific components. For example, graph attention alone results in only a ~2% improvement in accuracy on CIFAR-10 (Table 6), while weighted global pooling provides only a ~0.3% improvement on CIFAR-100 compared to standard global mean pooling. Clarifying the individual contributions of each of the four proposed components would make the paper more convincing to readers.

**Requested Changes:**

1. In Table 6, include a comparison for **GNN + Rehearsal Duplication (without attention)** to better isolate the contribution of the attention mechanism.

2. Include an ablation study demonstrating how hierarchical graph construction from multi-scale feature maps compares to alternative graph construction methods, such as SLIC-based superpixels or prior approaches like in DGN.

3. Clarify the individual contributions of each of the four components in the conclusion section to better align the claims with the empirical results.

---

> ### Author Response · Authors · 2026-05-19
> **Response to Reviewer tg3v**
>
> We thank the reviewer for the detailed reading of the paper and the provided feedback. We address the requested changes as follows:
>
> 1. We thank the reviewer for this helpful suggestion. We agree that including a “GNN + Rehearsal Duplication” variant without attention provides a cleaner isolation of the contribution of the attention mechanism within the proposed framework.
>
>    &nbsp;
>
>    To address this, we have extended Table 6 to include an additional ablation setting where rehearsal duplication is applied together with a non-attention GNN architecture. This enables a more direct comparison between graph-based relational learning with and without attention-based message passing under otherwise identical training conditions.
>
>    &nbsp;
>
>    The updated results show that while rehearsal duplication substantially improves performance overall, incorporating graph attention consistently provides additional gains across the evaluated datasets. This supports our hypothesis that attention-based neighborhood aggregation contributes beyond replay-related stabilization by enabling the model to selectively emphasize informative relational interactions during sequential learning.
>
>    &nbsp;
>
>    We have updated both the ablation table and the accompanying discussion in the manuscript accordingly.
>
>    &nbsp;
>
> 2. We thank the reviewer for this insightful suggestion and agree that comparisons against alternative graph construction strategies would further strengthen the analysis of the proposed framework.
>
>    &nbsp;
>
>    Our current work focuses specifically on hierarchical graph construction derived from multi-scale pretrained feature maps, as this formulation allows the graph representation to preserve semantic hierarchy across scales.
>
>    &nbsp;
>
>    We agree that evaluating alternative constructions, such as SLIC-based superpixels, would be valuable. However, integrating these methods into the current framework introduces several additional design considerations that would require careful and systematic treatment. In particular, SLIC-based methods produce graphs with dynamically varying node counts even for fixed-resolution images, which substantially affects the weighted pooling formulation and the subset-based weighing strategy proposed in Section 3.3. Furthermore, because superpixel regions are irregular and non-grid-aligned, defining consistent partitioning and positional correspondence for aggregation becomes non-trivial.
>
>    &nbsp;
>
>    Given these additional factors, we believe that a rigorous comparison would require a dedicated investigation to ensure fairness and avoid introducing confounding implementation choices. We therefore view this as an important direction for future work and have added discussion in the revised manuscript to motivate future exploration of alternative graph construction strategies within continual learning settings.
>
>    &nbsp;
>
> 3. We thank the reviewer for this helpful suggestion. We agree that the original conclusion could better distinguish the individual contributions of the proposed components and align the claims more precisely with the empirical findings from the ablation studies.
>
>    &nbsp;
>
>    In the revised conclusion, we have clarified the role of each major component separately: (1) the hierarchical graph construction for integrating multi-scale semantic representations, (2) the graph attention mechanism for relational message passing, (3) the weighted global pooling strategy for adaptive graph-level aggregation, and (4) the rehearsal duplication strategy for improving retention of past-task knowledge. In particular, we now explicitly note that rehearsal duplication contributes the largest quantitative gains in the ablation study, while the graph attention and relational representation components provide complementary improvements beyond replay-based stabilization alone.

---

### Review · Reviewer_exy9 · 2026-05-04

**Summary Of Contributions:**

The paper proposes ReReM, a framework for online continual learning in image classification that represents images as graphs and applies Graph Attention Networks over hierarchical features extracted from pretrained vision backbones. The method constructs graph representations from the images based on the internal ResNet18 or ViT features.  Then it uses GAT layers to model relations between spatial/semantic regions, and introduces a learnable weighted mean pooling mechanism for graph classification. The paper also proposes rehearsal duplication, where replay-buffer samples are repeated during training to rebalance past and current task data without increasing memory size. Experiments on Split CIFAR10, CIFAR100, and MiniImageNet report improved average accuracy and reduced forgetting compared to several CNN- and GNN-based continual learning baselines, with additional ablations for graph attention, rehearsal duplication, and pooling.

**Audience:**

Yes

**Audience Explanation:**

The paper addresses a relevant problem in continual learning and explores an interesting direction: using graph representations over pretrained visual features to mitigate forgetting. The idea of turning hierarchical feature maps into relational graphs and using GAT for online class-incremental learning may interest researchers working on continual learning, graph neural networks, and representation learning.

**Broader Impact Concerns:**

There is no evident concern about the paper's broader impact.

**Claims And Evidence:**

Yes

**Claims Explanation:**

The empirical results support that the proposed combination of pretrained hierarchical features, GAT-based graph processing, weighted pooling, and rehearsal duplication performs well on the benchmark datasets. The ablations are useful and show that rehearsal duplication in particular contributes substantially to the final performance.
However, the evidence does not fully isolate the contribution of the relational graph representation itself. The strongest gains appear after adding rehearsal duplication, and it is unclear how much of the improvement comes from the GNN representation versus the replay strategy. Some comparisons are also difficult to interpret because several baselines operate under different settings, use different backbones, or are taken from prior work rather than reproduced under a unified protocol. The ViT experiment is interesting, but comparing the method to offline baselines while emphasizing an online setting makes the evidence less clean.

**Requested Changes:**

1. Better isolate the contribution of the GNN component. Add comparisons where the same pretrained features, rehearsal buffer, LwF loss, and rehearsal duplication are used with a non-graph classifier or MLP baseline.
2. Clarify the role of rehearsal duplication. Since the largest gains in Table 6 appear after adding rehearsal duplication, provide stronger analysis of whether the improvement comes from relational representations or from increased replay exposure.
3. Clarify the ViT comparison. The paper should avoid comparing online ReReM too strongly against offline baselines, or should include online ViT-based baselines under the same setting.

---

> ### Author Response · Authors · 2026-05-19
> **Response to Reviewer exy9**
>
> We thank the reviewer for the thorough reading and detailed suggestions. We address the requested changes as follows:
>
> 1. We thank the reviewer for this valuable suggestion. We agree that isolating the contribution of the GNN component under matched training conditions is important for a fair assessment of the proposed framework.
>
>    &nbsp;
>
>    To address this concern, we have extended the ablation study to include a non-graph baseline in which the GATv2 layers are replaced with Linear layers while keeping the remaining components unchanged, including the pretrained feature extractor, rehearsal buffer, LwF objective, and rehearsal duplication strategy. This allows the comparison to isolate the effect of graph-based relational learning and attention-based message passing from the other continual learning components.
>
>    &nbsp;
>
>    The corresponding results are now included in Table 6 (“Baseline + R. Duplication”). We observe that while rehearsal duplication already provides strong improvements, the full graph-based formulation consistently achieves superior performance over the non-graph baseline, particularly on CIFAR100 and MiniImageNet. These findings suggest that the gains are not solely attributable to replay-related mechanisms, but also arise from the relational inductive bias introduced by the graph representation and attention-based neighborhood aggregation.
>
>    &nbsp;
>
>    We have revised the manuscript to clarify this experimental control and to better emphasize the role of the GNN component in the overall framework.
>
>    &nbsp;
>
> 2. We thank the reviewer for this important observation. We agree that the rehearsal duplication component contributes substantially to the final performance gains, and we have revised the discussion and conclusion to make this point more explicit.
>
>    &nbsp;
>
>    Importantly, our ablation results in Table 6 suggest that the improvement cannot be attributed solely to increased replay exposure. First, introducing graph attention alone (“GNN + G. Attn”) consistently improves performance over the base GNN across all datasets, indicating that attention based message passing itself provides a beneficial inductive bias for continual learning. Second, the full framework (“GNN + G. Attn + R. Duplication”) consistently outperforms both “GNN + R. Duplication” and “Baseline + R. Duplication”, demonstrating that rehearsal duplication and relational representations contribute complementary benefits rather than duplication alone accounting for the observed gains.
>
>
>    &nbsp;
>
>    Our interpretation is that rehearsal duplication primarily improves the retention strength of past-task information by increasing replay frequency, while the relational graph representations improve how this information is encoded and preserved through contextual interactions between semantic regions. In particular, the gains observed when moving from the non-attention GNN variants to the GAT-based variants suggest that attention-based relational updates help reduce interference during sequential learning beyond what replay exposure alone provides.
>
>    &nbsp;
>
>    To better clarify this distinction, we will revise the manuscript to explicitly discuss rehearsal duplication as an orthogonal stabilization mechanism that complements, rather than replaces, the contribution of relational representations in mitigating catastrophic forgetting.
>
>    &nbsp;
>
> 3. We thank the reviewer for this valuable feedback and agree that comparisons between online and offline continual learning settings should be interpreted cautiously due to the differing levels of difficulty and training constraints.
>
>    &nbsp;
>
>    Our intention in Section 4.9 was not to claim a direct state-of-the-art comparison against offline ViT-based methods, but rather to provide a preliminary investigation demonstrating that the proposed framework remains competitive even when extended to a ViT backbone and compared against less constrained settings. We acknowledge that the original wording may have overstated this comparison.
>
>    &nbsp;
>
>    To address this concern, we will revise the section to more clearly frame the ViT experiment as an exploratory generalization study rather than a direct benchmarking comparison. Specifically, we will soften the comparative claims and explicitly note that the cited baselines operate under offline rehearsal settings.
>
>    &nbsp;
>
>    We agree that inclusion of online ViT-based baselines under identical settings would provide a more rigorous comparison, and we will highlight this as an important direction for future work.

---

### Decision · Action_Editor_6AjT · 2026-06-12

**Recommendation:** Reject

**Additional Comments:**

A major revision should clarify the novelty of rehearsal duplication relative to existing replay strategies, better isolate the contribution of the graph representation under matched settings, report uncertainty for marginal gains, and avoid overstating comparisons across different continual learning settings.

I therefore recommend rejection in the current round, while encouraging a major revision and resubmission. A stronger version should substantially soften the claims, more clearly acknowledge rehearsal duplication as a main driver of performance, position it carefully with respect to existing rehearsal methods, and provide stronger matched evidence that relational graph representations offer a robust advantage in online continual learning.

**Audience:**

Yes

**Audience Explanation:**

The paper is relevant to TMLR, especially for researchers in continual learning, graph neural networks, and representation learning.

**Claims And Evidence:**

No

**Claims Explanation:**

The paper proposes ReReM, a framework for online continual learning in image classification that combines pretrained visual features, graph attention networks, weighted pooling, and rehearsal duplication. The reviewers found the topic relevant and the paper generally clear, and the revisions improved the component analysis.

The paper provides useful empirical results, but the evidence does not fully support the current framing. The revised ablations suggest that much of the performance improvement comes from rehearsal duplication, while the gains attributable specifically to the relational graph representation and graph attention components are comparatively modest. Since rehearsal duplication appears closely related to existing replay/rehearsal strategies, its novelty and distinction from prior work are not yet convincingly established. The claims should therefore be softened and better aligned with the evidence.

**Resubmission Of Major Revision:**

The authors may consider submitting a major revision at a later time.